# 3,3-Difluoroallyl ammonium salts: highly versatile, stable and selective *gem*-difluoroallylation reagents

Fei Ye[1,2,3], Yao Ge[2,3], Anke Spannenberg[2], Helfried Neumann [2], Li-Wen Xu [1] & Matthias Beller [2✉]

The selective synthesis of fluorinated organic molecules continues to be of major importance for the development of bioactive compounds (agrochemicals and pharmaceuticals) as well as unique materials. Among the established synthetic toolbox for incorporation of fluorine-containing units, efficient and general reagents for introducing -CF$_2$- groups have been largely neglected. Here, we present the synthesis of 3,3-difluoropropen-1-yl ammonium salts (DFPAs) as stable, and scalable *gem*-difluoromethylation reagents, which allow for the direct reaction with a wide range of fascinating nucleophiles. DFPAs smoothly react with *N*-, *O*-, *S*-, *Se*-, and *C*-nucleophiles under mild conditions without necessity of metal catalysts with exclusive regioselectivity. In this way, the presented reagents also permit the straightforward preparation of many analogues of existing pharmaceuticals.

[1] Key Laboratory of Organosilicon Chemistry and Material Technology of Ministry of Education, and Key Laboratory of Organosilicon Material Technology of Zhejiang Province, Hangzhou Normal University, No. 2318, Yuhangtang Road, Hangzhou 311121, PR China. [2] Leibniz-Institute for Catalysis, Albert-Einstein-Str. 29a, Rostock 18059, Germany. [3] These authors contributed equally: Fei Ye, Yao Ge. ✉email: matthias.beller@catalysis.de

In general, the substitution of C–H bonds by C–F bonds in organic compounds leads to significant changes in their physical, chemical, and biological behavior. More specifically, properties such as solubility, metabolic stability, hydrogen-bonding ability, lipophilicity, and chemical reactivity are strongly influenced by the incorporation of fluorine atoms[1–4]. Hence, an increasing number of products of the pharmaceutical and agrochemical industries as well as material sciences[5] contain fluorine atoms as a crucial part[6]. Indeed, nearly 20% of the top 200 drugs in 2016 belong to this class of compounds[7]. Due to the dynamic development in this area more than 40% (45% and 41%) of FDA-approved drugs in 2018[8], and 2019[9] are fluorine-containing. Even more pronounced, nearly 60% of the active ingredients for the agrochemical industry are fluorinated compounds. Not surprisingly, the incorporation of fluorine-containing groups has become one of the most important synthetic strategies for the design of new drugs and agrochemicals. Consequently, enormous efforts took place in academia and industry to develop such transformations[10] (Fig. 1a), and many prominent fluoroalkylating agents containing CF₃ or related groups were successfully introduced in the past years[11–15]. A selection of privileged reagents is shown in Fig. 1b.

In contrast to –CH₂F or –CF₃ substituents, –CF₂–R containing bioactive compounds are significantly less explored and comparably few marketed therapeutic agents exist yet (for some examples see Fig. 1c)[16–23]. So far, the most common route for their synthesis makes use of the fluorination of a pre-existing functional group using for example DAST or Deoxfluor[24–26]. Apart from the hazardous nature of these reagents, this approach is also limited with respect to functional group compatibility.

In our opinion, the restricted availability of general and selective synthetic methodologies clearly impeded the development of versatile building blocks with functionalized difluoromethyl units. As an alternative strategy, the direct introduction of difluoroalkyl groups forming X−CF₂−R bonds (X = C, O, N, S, etc.; R = C) could solve this problem[27–33]. Especially, using *gem*-difluoroallyl moieties (a –CF₂– group neighboring to an alkene unit) offers attractive possibilities because of the wide variety of subsequent functionalizations and its occurrence in existing pharmaceuticals (see also Fig. 1c)[16–20]. Unfortunately, so far only a few methodologies for the direct intermolecular preparation of *gem*-difluoroallylic products are known[34–46]. This can be explained by the lack of suitable reagents like 3,3-difluoroallyl-X compounds[47,48]. Thus, 3-bromo-3,3-difluoropropene (BDFP) has become the most popular surrogate for such transformations[34–36,47], which has been applied exemplarily to construct difluoroallylated alcohols via metal–halogen exchange reactions (Fig. 1d, left)[47–50]. In addition, Tsuji–Trost allylations of BDFP and related substrates were also disclosed (Fig. 1d, right)[34–38]. More specifically, the groups of Zhang[34], Ichikawa[35], and most recently Cogwell[36], reported interesting Pd-catalyzed *gem*-difluoroallylation of organoboranes, halophenols, and anilines, respectively.

Nevertheless, the use of BDFP has limitations due to its physical properties, e.g. volatility (b.p.: 41–42 °C) and substrate scope. In addition, the regioselective synthesis of the desired *gem*-difluoroallylated products requires the aid of stoichiometric amounts of metals or precious catalysts. In conclusion, to date, a general approach that meets the criteria of both broad reaction scope and high selectivity for *gem*-difluoroallylation reactions remains elusive.

As a solution, herein we describe the development of a family of easily available and bench-stable *gem*-difluorinated allyl ammonium salts (DFAAs), which allow exclusive γ-regioselective *gem*-difluoroallylation of *O*-, *N*-, *S*-, *Se*-, and *C*-nucleophiles (Fig. 1e). The resulting products, featuring a synthetically highly valuable Nu−CF₂CH=CH₂ unit, allow for the straightforward synthesis of diverse organofluorine compound libraries that offer numerous possibilities to enlarge the chemical space in the development of bioactive molecules.

## Results and discussion

Although quaternary allyl ammonium salts are known for more than a century[51], they received surprisingly little attention[52–55]. While investigating palladium-catalyzed cascade processes for the synthesis of spiro-fused heterocycles[56], recently we noticed the utility and advantages of this class of compounds compared to other allylic electrophiles such acetates, halides, and so on. Inspired by this work and our interest in organofluorine building blocks[56–61], we became interested in the synthesis and applications of DFAAs. Surprisingly, the parent 3,3-difluoropropen-1-yl ammonium salts (DFPAs) have not been described before.

Starting from 3,3,3-trifluoropropene, which is a commercial raw material, which is used for the preparation of a variety of products (e.g. in cosmetics, chemical manufacturing, etc.), base-mediated amination with piperidine provided γ,γ-difluoroallyl piperidine smoothly as a colorless oil. Subsequent selective *N*-methylation[38], gave the desired 3,3-difluoropropen-1-yl ammonium salt **1a** as a crystalline white powder without the need for chromatography or other tedious purifications in 56% overall yield. Following this two-step protocol other γ,γ-fluoro-substituted derivatives **1b–1e** can be obtained on gram-scale in a straightforward manner (Fig. 2a). Notably, all resulting products can be conveniently handled and are air-stable for months.

Having a small representative selection of fluorinated allyl ammonium salts in hand, we were interested to investigate their general reactivity. Here, we were especially interested in allylic substitution reactions with oxygen-derived or nitrogen-derived nucleophiles because the resulting structural motifs RCF₂−O, RCF₂−N are not easily accessible by other synthetic means. Thus, the model reaction of γ,γ-difluoroallyl ammonium salt **1a** with 4-phenyl phenol **2** was performed in the presence of a palladium catalyst under typical Tsuji–Trost conditions (toluene, 80 °C, PdBr₂, *n*-butyl-diadamantylphosphine, cesium carbonate)[62]. To our delight, the desired difluoroallylic aryl ether product **3** is obtained in 92% yield and high selectivity (>99/1%; Fig. 2b, entry 1). Interestingly, performing a control experiment under identical conditions without the palladium catalyst revealed similar product yields (87% yield; Fig. 2b, entry 2), which can be explained by intrinsic reactivity for direct SN2′ substitution in the presence of a base. Notably, even without any precious metal catalyst present, exclusive γ-selective substitution (>99/1) was observed without any detectable formation of the α-substitution product. In contrast, for other allylic electrophiles such behavior in the absence of transition metal catalysts is difficult to control[63].

Next, to obtain optimal results for this benchmark system, critical reaction parameters were studied in detail (see Supplementary Tables 1–3). The following observations are important to note: (a) It is possible to run the nucleophilic substitution efficiently under very mild conditions (RT to 50 °C) in the presence of sub-stoichiometric amounts of base without any metal catalyst (Fig. 2b, entries 3–10); (b) this transformation works well in aqueous solutions as well as pure water and high yields of **3** (up to 94%) can be obtained, which is interesting for biologically relevant substrates and considering the importance of green solvents (Fig. 2b, entries 6–11)[64]; (c) using stronger base (sodium hydride), the reaction proceeded smoothly within minutes (typically <10 min) at ambient temperature affording the desired ether product **3** in quantitative yield (Fig. 2b, entry 12). Interestingly, depending on the physical properties of specific substrates, the most suited protocol might be selected from these orthogonal approaches. Similar results were obtained using other

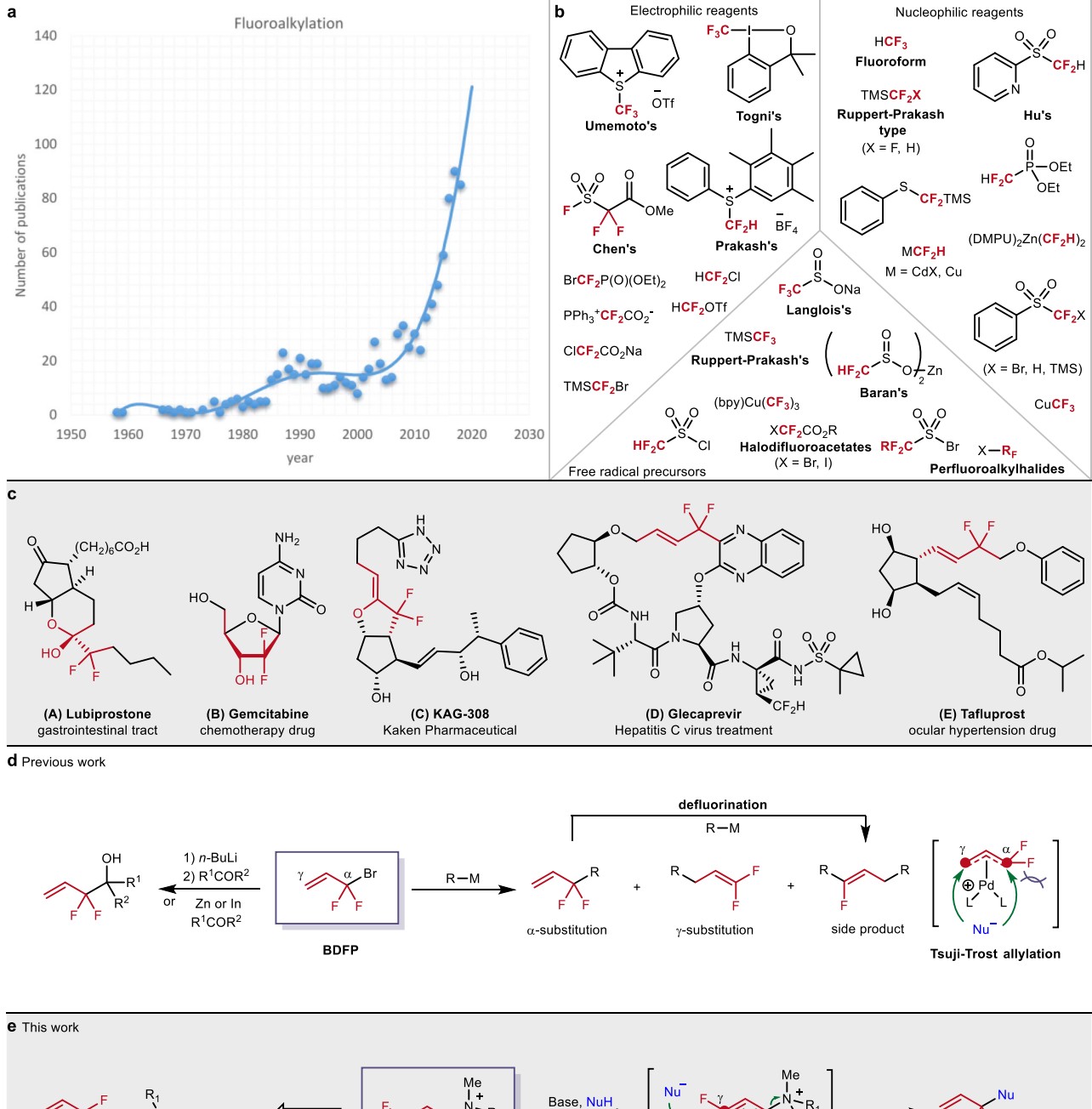

**Fig. 1 The allure of fluoroalkylations. a** The dynamic development of fluoroalkylations in the past years. **b** Commonly used fluoroalkyl-containing reagents for direct preparation of fluoroalkyl compounds. **c** The importance of compounds containing difluoromethylene unit. **d** The challenge of the direct preparation of *gem*-difluoroallyl compounds using BDFP. **e** The presented strategy for the selective construction of *gem*-difluoroallyl compounds.

related ammonium salts **1b–1e** as difluoroallylating reagents (Fig. 2b, entries 13–16).

After having demonstrated the efficiency of the model system at different conditions, we were interested in the general reaction scope. At first, other oxygen nucleophiles were studied. Using NaH as base, the *gem*-difluoroallylation of allylic ammonium salt **1a** with various phenols provided allylic aryl ethers **3–18** in good

to excellent yields. The reaction displayed remarkable compatibility with functional groups including ketones, aldehydes, esters, halides, amines, boronic acidic esters, heterocycles, and alkenes. As shown in the case of product **19**, the reaction can be easily run on gram-scale. Apart from phenols, a variety of aliphatic alcohols were successfully examined as well, affording the allylic alkyl ethers **23–27** in 63–99% yields. From a synthetic point of view, it

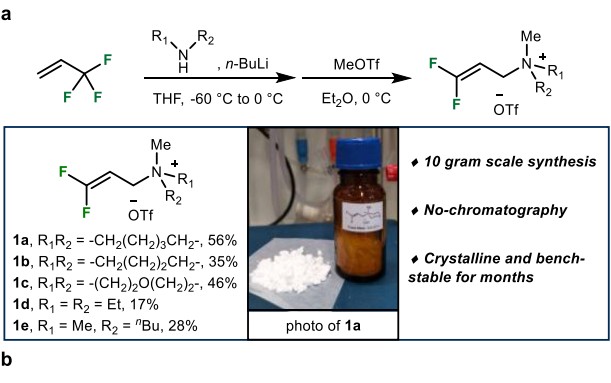

**a**

**b**

| Entry | 1 | Base (x equiv) | Solv. (con.) | T (°C) | t (h) | yield (%) |
|---|---|---|---|---|---|---|
| 1† | 1a | $Cs_2CO_3$ (1.0) | Toluene (0.1 M) | 80 | 18 | 92 |
| 2 | 1a | $Cs_2CO_3$ (1.0) | Toluene (0.1 M) | 80 | 18 | 87 |
| 3 | 1a | $Cs_2CO_3$ (0.5) | Toluene (0.1 M) | 80 | 18 | 96 |
| 4 | 1a | $Cs_2CO_3$ (0.5) | Toluene (0.1 M) | 50 | 18 | >99 |
| 5 | 1a | $Cs_2CO_3$ (0.2) | Toluene (0.1 M) | 80 | 18 | 59 |
| 6 | 1a | $K_2CO_3$ (1.0) | $H_2O$/THF (9:1, 0.1 M) | RT | 18 | 49 |
| 7 | 1a | KOH (1.0) | $H_2O$/THF (9:1, 0.1 M) | RT | 18 | 72 |
| 8 | 1a | NaOH (1.0) | $H_2O$/THF (9:1, 0.1 M) | RT | 18 | 64 |
| 9 | 1a | $Cs_2CO_3$ (1.0) | $H_2O$/THF (9:1, 0.1 M) | RT | 18 | 63 |
| 10 | 1a | $Cs_2CO_3$ (0.5) | $H_2O$/THF (9:1, 0.1 M) | 80 | 18 | 94 |
| 11 | 1a | $Cs_2CO_3$ (0.5) | $H_2O$ (0.1 M) | 80 | 18 | 90 |
| 12‡ | 1a | NaH (1.5) | DMF (0.2 M) | RT | 0.5 | >99 |
| 13‡ | 1b | NaH (1.5) | DMF (0.2 M) | RT | 0.5 | 97 |
| 14‡ | 1c | NaH (1.5) | DMF (0.2 M) | RT | 0.5 | 97 |
| 15‡ | 1d | NaH (1.5) | DMF (0.2 M) | RT | 0.5 | 98 |
| 16‡ | 1e | NaH (1.5) | DMF (0.2 M) | RT | 0.5 | 90 |

**Fig. 2 Reagent synthesis and model reaction. a** Two-step synthesis procedure for the preparation of difluorinated allyl ammonium salts **1a–1e**. **b** Optimization studies for the $S_N2'$ substitution reaction of difluoroallyl ammonium salt **1a** with 4-phenyl phenol **2** in the presence of different basic conditions. †PdBr$_2$ (0.015 mmol) and $^n$BuPAd$_2$ (0.03 mmol) were used. ‡The reaction was performed under an argon atmosphere.

is also interesting that *gem*-difluoroallylation of functionalized propargylic alcohols generated versatile diene-ynes (**25** and **26**) and 1,6-enyne (**27**). Notably, less common oxygen nucleophiles such as oximes and hydroxylamines also worked well and provided the respective products **20–22** in very good yields. Recently, these latter products attracted attention because of their potential application for fluoroalkoxylation of alkyl halides[65].

In addition to phenols, aliphatic and propargylic alcohols, also diverse sulfur-, selenium-, nitrogen-, as well as activated carbon-nucleophiles can be used according to our general protocol as depicted in Fig. 3. More specifically, aryl thiols bearing electron-donating or electron-withdrawing groups and alkyl thiols as well as benzeneselenol underwent the desired the *S*- or *Se*-*gem*-difluoroallylation leading to the corresponding products **28–32**. Gratifyingly, these reaction conditions can be used for a broad range of heterocyclic amines and tosyl-protected secondary

amines, too. In all cases, the reaction proceeded smoothly to give **33–38** bearing allylic *N*-difluoromethylene unit in high yields.

Next, we turned our attention to the reaction of activated carbon nucleophiles. Interestingly, using ketones with β-H resulted selectively in *O*-*gem*-difluoroallylation and provided the corresponding allyl vinyl ethers **39–42** as single products in 73–99% yields. The high carbon/oxygen (C/O) regioselectivity could be explained by the Pearson acid–base concept[66,67]. On the other hand, using diphenylacetonitrile led to the construction of $C_{sp3}$–$CF_2R$ bond (**43**) and thus to access α-difluoroalkylated carboxylic acid derivatives. It is important mentioning that in all cases, the α/γ-regioselective *gem*-difluoroallylation undoubtedly generated exclusively the γ-substitution product. At this point it should be also noted that many difluoroallyl ammonium salts can be used for the preparation of more substituted derivatives following a similar methodology (see Supplementary Fig. 3).

To showcase the value of this synthetic approach, late-stage functionalization of a variety of pharmaceuticals, bioactive molecules, and natural products are depicted in Fig. 4. More specifically, estrone, one of the three major endogenous estrogens, with a phenolic group delivered to the difluoroallylic derivative **44** in excellent yield (97%). The molecular structure of **44** was confirmed by X-ray crystallography. Similar transformations performed with the (+)-α-tocopherol, acetate-protected β-estra-diol, and diethylstilbestrol afforded the desired products **45–47** with high efficiency (99%, 97%, and 92%, respectively). Further-more, monoterpene (*Z*)-nerol—known as sweet rose-like aroma—bearing an allylic hydroxyl group, gave the corresponding 1,6,10-triene **48** in 97% yield. Additionally, several functionalized biologically relevant primary and secondary alcohols and various pharmaceutically important molecules, such as mestranol, lynestrenol, quinine, and testosterone, containing sterically con-gested tertiary or secondary hydroxyl groups, underwent this reaction smoothly, giving **49–55** in good to excellent yields. Notably, a derivative of the sulfonamide Celecoxib (analgesic) also could be coupled to give the *N*-difluoroallylated product **56** in high yields.

Difluoroallylic $S_N2'$ substitution of different substrates con-taining more than one activated proton further highlights the importance of this method, especially in the late-stage functio-nalization of bioactive molecules. To ensure chemoselectivity, the milder base $Cs_2CO_3$ was applied in these examples. For example, ezetimibe, an FDA-approved drug for the treatment of high blood cholesterol and certain other lipid abnormalities, underwent selective *gem*-difluoroallylation to give **60** in 84% yield. This demonstrates the compatibility of the reaction conditions with aliphatic hydroxyl groups and an amide-activated β-protons. In addition, transformations of *N*-Boc-protected *L*-tyrosine deriva-tive, ethynylestradiol, and β-estradiol afforded the desired pro-ducts **57–59** in high yields.

After validating the generality of our protocol and having a selection of allylic *gem*-difluoride building blocks in hand, we wanted to demonstrate their value by further valorizations (Fig. 5). Exemplarily, we first carried out the Pd-catalyzed alkoxycarbonylation of *gem*-difluoroallylic aryl ether **3** without further optimizations. Taking advantage of the specific phosphine ligand **L1** with built-in-base functions[68,69], the corresponding product **61** was obtained in 73% yield with high regioselectivity for the linear product. Furthermore, under-reported reaction conditions[70], **3** underwent highly regioselective hydroformylation using simple $Co_2(CO)_8$ as the pre-catalyst (87% yield, *l/b* > 20/1).

Apart from aldehydes and carboxylic acids, the difluorinated products constitute virtuous precursors for the synthesis of other multifunctional compounds. For example, dihydroxylation of **16** and **23** provided the corresponding difluoromethylated glycerol derivatives **63** and **64** in quantitative yields (Fig. 5b). These

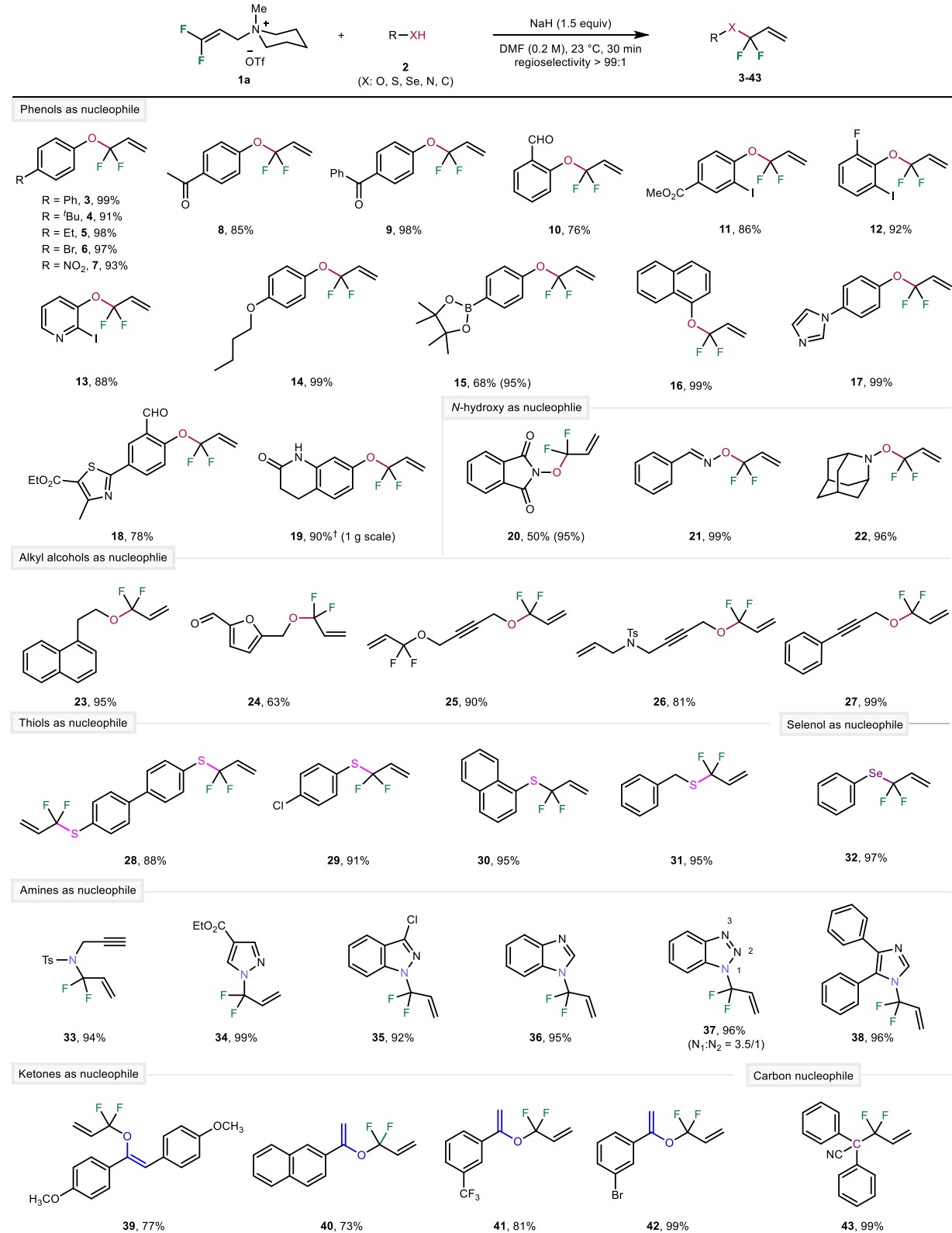

**Fig. 3 Regioselective substitution of 1a by different O-, N-, S-, Se- and C-nucleophiles: Substrate scope.** Standard reaction conditions: **1** (1.2 equiv), **2** (1 equiv), NaH (1.5 equiv) in DMF (0.2 M), the reaction mixture was performed at 23 °C under argon atmosphere for 30 min, isolated yield, the α/γ regioselectivity was determined by crude $^{19}$F NMR analyses. $^{†}$Cs$_2$CO$_3$ (0.5 equiv) and toluene (0.1 M) were used as base and solvent, the reaction was performed at 50 °C under air atmosphere for 18 h.

**Fig. 4 Selective 1,1-difluoroallylation of natural products and pharmaceuticals.** Standard reaction conditions: **1a** (1.2 equiv), **2** (1 equiv), NaH (1.5 equiv) in DMF (0.2 M), the reaction mixture was performed at 23 °C under argon atmosphere for 30 min, isolated yield, the α/γ regioselectivity was determined by crude $^{19}$F NMR analyses. $^†$Cs$_2$CO$_3$ (0.5 equiv) and toluene (0.1 M) were used as base and solvent, the reaction was performed at 50 °C under air atmosphere for 18 h.

compounds are of interest as potential precursors for the preparation of difluorinated propranolol and many other analogs of beta blocker pharmaceuticals[71].

Considering transition-metal-catalyzed carbocycloaddition reactions as powerful methods for the construction of polyheterocycles[72], products **26** and **27** were examined following previously reported protocols. Indeed, starting from diene-yne **26**, an intramolecular ruthenium-catalyzed [2 + 2 + 2] co-cyclization took place to yield the N,O-fused tricyclic heterocycle **65** in 69% yield, albeit with moderate diastereoselectivity. The asymmetric version of rhodium-catalyzed reductive cyclization of 1,6-enyne **27** successfully proceeded at ambient temperature and hydrogen pressure, affording the *gem*-difluorinated alkylidene-substituted

furan **66** in excellent yield and high enantioselectivity (97% yield, 92% *ee*).

Finally, two *gem*-difluoromethylene-containing analogs of actual drug compounds were prepared using the here presented methodology. Aripiprazole is an antipsychotic drug, ranking 128th among the top 200 retail drugs in 2019 with annual sales >1.2 billion dollars[73]. In a straightforward manner the O–CF$_2$-derivative of aripiprazole was successfully synthesized from commercially available materials, as illustrated in Fig. 5e. After highly regioselective hydroformylation of difluoroallylic aryl ether **19**, followed by subsequent reduction, mesylation, and final S$_N$2 substitution, the desired difluorinated aripiprazole **70** was obtained in 68% overall yield. As the second example, the

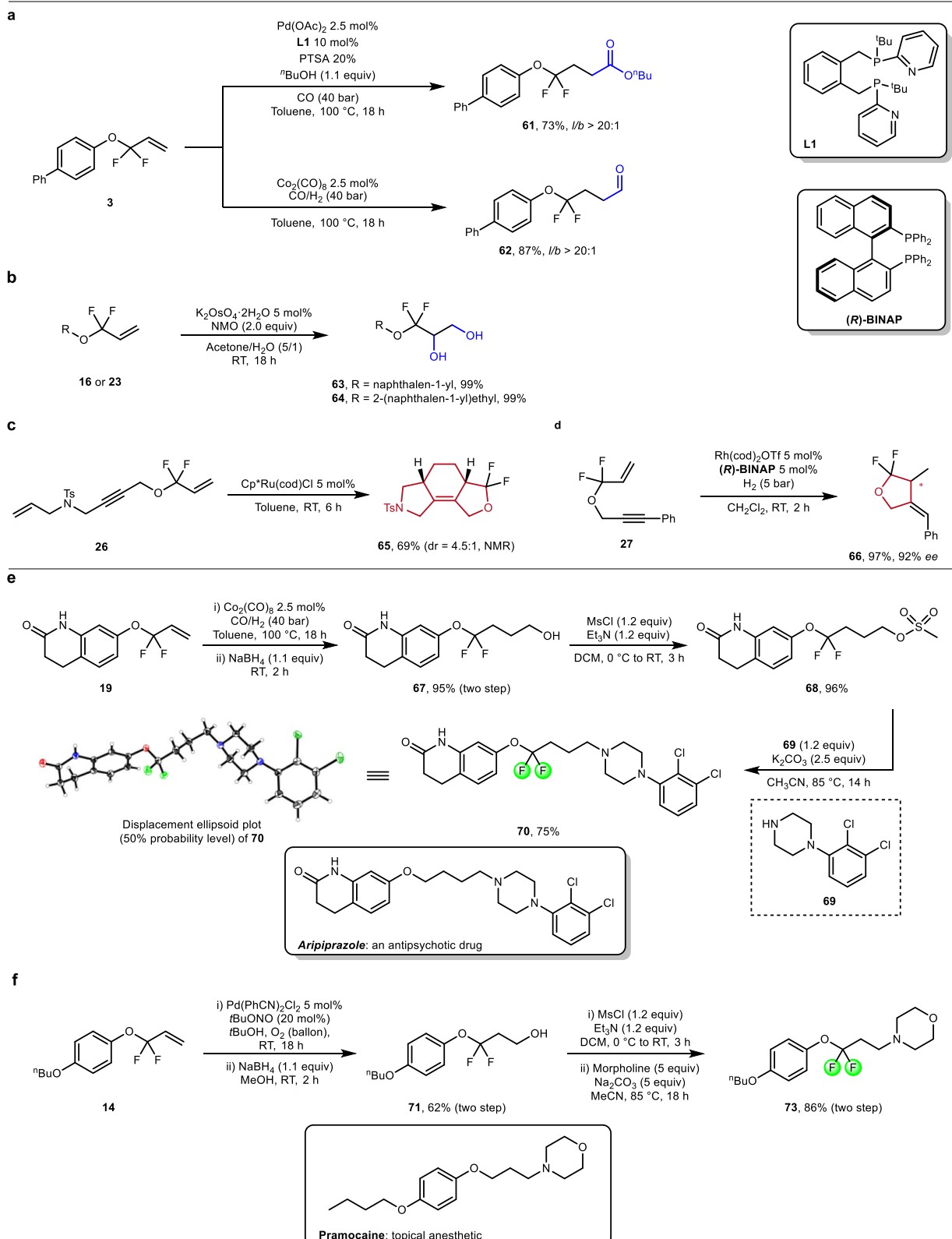

**Fig. 5 Synthesis of *gem*-difluorinated analogs of drugs and further valorization of *gem*-difluorinated allyl ethers. a** Transition-metal catalyzed carbonylation of difluorinated alkene **3**. **b** Dihydroxylation of **16** and **23** to give difluorinated 1-aryl-glycerols. **c** Ruthenium-catalyzed [2 + 2 + 2] co-cyclization of diene-yne **26**. **d** Rhodium-catalyzed reductive cyclization of 1,6-enyne **25**. **e** Practical procedures for the synthesis of *OCF₂*-Aripiprazole **70**. **f** Four-step protocol for the synthesis of *OCF₂*-Pramocaine **73**.

$CF_2$-analog of Pramocaine (a topical anesthetic) was prepared easily in four steps in 53% overall yield (Fig. 5f).

In summary, we present the parent examples of DFPAs, which can be easily prepared by base-mediated amination of the industrial feedstock 3,3,3-trifluoropropene. The resulting compounds are air-stable and water-stable and can be conveniently applied as highly efficient difluoromethylene reagents. Following an operationally simple and scalable protocol various nucleophiles including phenols, alcohols, thiols, selenols, amines, ketones, and alkanes with activated $C_{sp3}$-H undergo exclusively γ-regioselectivity gem-difluoroallylation without the need of transition-metals under mild conditions. So far, the resulting allylic gem-difluoromethylated products were difficult to access by using conventional approaches. The successful incorporation of the allylic difluoromethylene unit into a wide variety of bioactive pharmaceuticals and natural products provides chemists a new route to explore fluorine-containing molecular scaffolds.

## Methods

**General procedure A for the preparation of allylic gem-difluoromethylated compounds**. To a 25 mL oven-dried pressure tube equipped with a magnetic stir bar were added difluoroallyl ammonium salt **1** (0.36 mmol), NaH (0.45 mmol), DMF (1.5 mL), and then nucleophiles **2** (0.3 mmol) was introduced under argon atmosphere. The sealed pressure tube was vigorously stirred at 23 °C for 30 min. The reaction mixture was quenched with few drops of water and extracted with ethyl acetate (3 × 20 mL). The combined organic layer was dried over anhydrous $Na_2SO_4$, filtered and concentrated. The residue was purified by chromatography on silica gel (It is worthy to note that some of the fluorinated product can only be separated without decomposition with 2% triethylamine in the eluent) to afford the desired product.

**General procedure B for the preparation of allylic gem-difluoromethylated compounds**. To a pressure tube (25 mL) equipped with a magnetic stir bar difluoroallyl ammonium salt **1** (0.36 mmol), $Cs_2CO_3$ (0.15 mmol), toluene (2.5 mL), and then nucleophiles **2** (0.3 mmol) were added. The sealed pressure tube was vigorously stirred for 18 h at 50 °C under air atmosphere. After cooling to r.t., the reaction mixture was diluted with ethyl acetate (10 mL) and filtered through a short pad of celite eluting with ethyl acetate (3 × 10 mL). After evaporation, the residue was purified by chromatography on silica gel to afford the desired product. Notably, some of the fluorinated products can only be separated without decomposition with 2% triethylamine in the eluent.

## Data availability

The authors declare that all the data supporting this study, including the experimental details, data analysis, and spectra for all unknown compounds, see Supplementary Files. All data underlying the findings of this work are available from the corresponding author upon reasonable request. The X-ray crystallographic coordinates for structures reported in this study have been deposited at the Cambridge Crystallographic Data Centre (CCDC), under deposition numbers 2035829 (**44**) and 2035828 (**70**). These data are provided free of charge by the joint Cambridge Crystallographic Data Centre and Fachinformationszentrum Karlsruhe Access Structures service www.ccdc.cam.ac.uk/structures.

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

# ARTICLE

38. Tang, L., Liu, Z.-Y., She, W. & Feng, C. Selective single C–F bond arylation of trifluoromethylalkene derivatives. *Chem. Sci.* **10**, 8701–8705 (2019).

39. Belhomme, M.-C., Poisson, T. & Pannecoucke, X. Copper catalyzed α-difluoroacetylation of dihydropyrans and glycals by means of direct C–H functionalization. *Org. Lett.* **15**, 3428–3431 (2013).

40. Sato, K., Omote, M., Ando, A. & Kumadaki, I. Reactions of ethyl bromodifluoroacetate in the presence of copper powder. *J. Fluor. Chem.* **125**, 509–515 (2004).

41. Schwaebe, M. K., McCarthy, J. R. & Whitten, J. P. Nickel(0)-catalyzed coupling of vinylzirconiums to α-bromo-αα-difluoro esters. Convenient generation of a functionalized allyldifluoro moiety. *Tetrahedron Lett.* **41**, 791–794 (2000).

42. Li, G. et al. Nickel-catalyzed decarboxylative difluoroalkylation of α,α-unsaturated carboxylic acids. *Angew. Chem. Int. Ed.* **55**, 3491–3495 (2016).

43. Wu, G. & Wangelin, A. J. Stereoselective cobalt-catalyzed halofluoroalkylation of alkynes. *Chem. Sci.* **9**, 1795–1802 (2018).

44. Feng, Z., Min, Q.-Q., Zhao, H.-Y., Gu, J.-W. & Zhang, X. A general synthesis of fluoroalkylated alkenes by palladium-catalyzed Heck-type reaction of fluoroalkyl bromides. *Angew. Chem. Int. Ed.* **54**, 1270–1274 (2015).

45. Guo, W.-H., Zhao, H.-Y., Luo, Z.-J., Zhang, S. & Zhang, X. Fluoroalkylation −borylation of alkynes: an efficient method to obtain (Z)-tri- and tetrasubstituted fluoroalkylated alkenylboronates. *ACS Catal.* **9**, 38–43 (2019).

46. Li, Z., Cui, Z. & Liu, Z.-Q. Copper- and iron-catalyzed decarboxylative tri- and difluoromethylation of α,α-unsaturated carboxylic acids with CF3SO2Na and (CF2HSO2)2Zn via a radical process. *Org. Lett.* **15**, 406–409 (2013).

47. Qing, F.-L. *3-Bromo-3,3-Difluoropropene. e-EROS Encylcopedia of Orgnic Synthesis* (John Wiley & Sons, Ltd., 2005).

48. Tellier, F., Duffault, J.-M., Baudry, M. & Sauvêtre, R. Reactivity of 1-bromo-1,1-difluoro-2-alkenes synthesis of 1,1-difluoroolefins. *J. Fluor. Chem.* **91**, 133–139 (1998).

49. Yue, X., Qiu, X. & Qing, F. Metal-mediated gem-difluoroallylation of N-acylhydrazones: highly efficient synthesis of α,α-difluorohomoallylic amines. *Chin. J. Chem.* **27**, 141–150 (2009).

50. Seyferth, D., Simon, R. M., Sepelak, D. J. & Klein, H. A. gem-(Difluoroallyl) lithium: preparation by lithium-halogen exchange and utilization in organosilicon and organic synthesis. *J. Am. Chem. Soc.* **105**, 4634–4639 (1983).

51. Menschutkin, N. Beiträge zur Kenntnis der Affinitätskoeffizienten der Alkylhaloide und der organischen Amine. *Z. Phys. Chem.* **5**, 589–600 (1890).

52. Soheili, A. & Tambar, U. K. Tandem catalytic allylic amination and [2,3]-Stevens rearrangement of tertiary amines. *J. Am. Chem. Soc.* **133**, 12956–12959 (2011).

53. West, T. H., Daniels, D. S., Slawin, A. M. & Smith, A. D. An isothiourea-catalyzed asymmetric [2,3]-rearrangement of allylic ammonium ylides. *J. Am. Chem. Soc.* **136**, 4476–4479 (2014).

54. Arfaoui, A., Saâdi, F., Nefzi, A. & Amri, H. Easy conversion of dimethylα-(bromomethyl)fumarate into functionalized allyl ethers mediated by DABCO. *Synth. Commun.* **45**, 2627–2635 (2015).

55. Baidya, M., Remennikov, G. Y., Mayer, P. & Mayr, H. SN2' versus SN2 reactivity: control of regioselectivity in conversions of Baylis–Hillman adducts. *Chem. Eur. J.* **16**, 1365–1371 (2010).

56. Ye, F., Ge, Y., Spannenberg, A. H., Neumann & Beller, M. The role of allyl ammonium salts in palladium-catalyzed cascade reactions towards the synthesis of spiro-fused heterocycles. *Nat. Commun.* **11**, 5383 (2020).

57. He, L. et al. Heterogeneous platinum-catalyzed C–H perfluoroalkylation of arenes and heteroarenes. *Angew. Chem. Int. Ed.* **54**, 4320–4324 (2015).

58. Natte, K. et al. Palladium-catalyzed trifluoromethylation of (hetero)arenes with CF3Br. *Angew. Chem. Int. Ed.* **55**, 2782–2786 (2016).

59. Zhang, S. et al. A general and practical Ni-catalyzed C–H perfluoroalkylation of (hetero)arenes. *Chem. Commun.* **55**, 6723–6726 (2019).

60. Ye, F. et al. Versatile fluorinated building blocks by stereoselective (per) fluoroalkenylation of ketones. *Eur. J. Org. Chem.* **2020**, 70–81 (2020).

61. Zhang, S. et al. Selective nickel-catalyzed fluoroalkylations of olefins. *Chem. Commun.* **56**, 15157–15160 (2020).

62. Trost, B. M. New rules of selectivity: allylic alkylations catalyzed by palladium. *Acc. Chem. Res.* **13**, 385–393 (1995).

63. Fujita, T., Sanada, S., Chiba, Y., Sugiyama, K. & Ichikawa, J. Two-step synthesis of difluoromethyl-substituted 2,3-dihydrobenzoheteroles. *Org. Lett.* **16**, 1398–1401 (2014).

64. Song, H.-X., Han, Q.-Y., Zhao, C.-L. & Zhang, C.-P. Fluoroalkylation reactions in aqueous media: a review. *Green Chem.* **20**, 1662–1731 (2018).

65. Li, Y., Yang, Y., Xin, J. & Tang, P. Nucleophilic trifluoromethoxylation of alkyl halides without silver. *Nat. Commun.* **11**, 755 (2020).

66. Mayr, H., Breugst, M. & Ofial, A. R. Farewell to the HSAB treatment of ambient reactivity. *Angew. Chem. Int. Ed.* **50**, 6470–6505 (2011).

67. Ho, T.-L. The hard soft acids bases (HSAB) principle and organic chemistry. *Chem. Rev.* **75**, 1–20 (1975).

68. Dong, K. et al. Efficient palladium-catalyzed alkoxycarbonylation of bulk industrial olefins using ferrocenyl phosphine ligands. *Angew. Chem. Int. Ed.* **56**, 5267–5271 (2017).

69. Liu, J., Yang, J., Ferretti, F., Jackstell, R. & Beller, M. Pd-catalyzed selective carbonylation of gem-difluoroalkenes: a practical synthesis of difluoromethylated esters. *Angew. Chem. Int. Ed.* **58**, 4690–4694 (2019).

70. Fanfonia, L., Diaba, L., Smejkal, T. & Breit, B. Efficient synthesis of new fluorinated building blocks by means of hydroformylation. *CHIMIA* **68**, 371–377 (2014).

71. Wang, Z.-Y., Wang, Y., Sun, L.-W. & Zhu, J.-T. Asymmetric synthesis of (R)- and (S)-Moprolol. *Chem. Res. Chin. Univ.* **24**, 747–751 (2008).

72. Nakamura, I. & Yamamoto, Y. Transition-metal-catalyzed reactions in heterocyclic synthesis. *Chem. Rev.* **104**, 2127–2198 (2004).

73. The Njarðarson group. *Top 200 Pharmaceuticals by Retail Sales in 2019.* https://njardarson.lab.arizona.edu/sites/njardarson.lab.arizona.edu/files/Top%20200%20Drugs%20By%20Retail%20Sales%20in%202019V2.pdf (2019).

## Acknowledgements

We are grateful for financial support from the State of Mecklenburg-Western Pomerania and the Federal State of Germany (BMBF). F.Y. and L.-W.X. thank the National Natural Science Foundation of China (Nos. 21773051, 22072035, and 21801056) for financial support. We also thank the Analytic Department (LIKAT) for their kind support.

## Author contributions

M.B. and F.Y. conceived and designed the experiments. F.Y. and Y.G. performed the experiments and analyzed the data. A.S. performed the X-ray analysis. H.N. and L.-W.X. participated in the discussions and supported the project. M.B. and F.Y. prepared the manuscript with feedback from all authors.

## Funding

## Competing interests

The authors declare no competing interests.
