## [Peer Review File · Nature Communications]

Reviewers' Comments:

Reviewer #1:

Remarks to the Author:

This manuscript by Beller and coworkers reports the synthesis and application of a new class of reagents (3,3-difluoroallyl ammonium salts, DFPA) for the difluoro-allylation of N-, O-, S-, Se- and C-centered nucleophiles. As I stated in the previous comments, this work is well-done and will offer a new method to the synthesis of gem-difluoro compounds. The revised manuscript addressed the issues I raised in the previous submission. Therefore, I recommend the publication of this work on *Nat. Commun.*

Some minor issues:

- 1) The references for the FNMR (or if not referenced) should be mentioned in the General Information of the supporting information.
- 2) The authors mentioned in the main text that "some of the fluorinated product can only be separated without decomposition with 2% triethylamine in the eluent". Yet, none the eluent contains triethylamine in the supporting information. The authors should clarify this point in the SI so that the separation could be reproduced in other labs.
- 3) The authors have polished the language in the main text, there are still many grammatical issues in the supporting information.

Reviewer #2:

Remarks to the Author:

The current paper discloses a useful fluoroalkylating reagent 3,3-difluoropropen-1-yl ammonium salts (DFPAs) for preparation of gem-difluoroallylated compounds. This reagent can be readily prepared from industry chemical 3,3,3-trifluoropropene and allows gem-difluoroallylation of a variety of O-, N-, S-, Se- nucleophiles. The resulting products can serve as a versatile building block for diversified synthesis of fluorinated compounds of medicinal interest. However, the C-nucleophiles seem to be improved their substrate scope. In the current process, only one specific substrate diphenylacetonitrile was applicable to the reaction. Reactions of ketones with beta-H resulted in gem-difluoroallyl vinyl ethers. Although the SN2' reaction used in this paper is well established, the reagent enables to access a variety of gem-difluoroallylated compounds that are not easy to prepare via transition-metal free conditions. Overall, this is a nice paper, I recommend publication of the paper and suggest minor changes:

- 1) "difluoroallylation" should be "gem-difluoroallylation".
- 2) Explanations should be provided for the formation of gem-difluoroallyl vinyl ethers with ketones bearing beta-H as the substrates.
- 3) Cite the following papers for the direct introduction of difluoroalkyl groups forming X-CF₂-R: *Acc. Chem. Res.* 2018, 51, 2264; *Nat. Chem.* 2017, 9, 918; *Nat. Chem.* 2019, 11, 948.

Reviewer #3:

None

REVIEWER COMMENTS

Reviewer #1 (Remarks to the Author):

This manuscript by Beller and coworkers reports the synthesis and application of a new class of reagents (3, 3-difluoroall ammonium salts, DFPA) for the difluoro-allylation of N-, O-, S-, Se- and C-centered nucleophiles. As I stated in the previous comments, this work is well-done and will offer a new method to the synthesis of gem-difluoro compounds. The revised manuscript addressed the issues I raised in the previous submission. Therefore, I recommend the publication of this work on Nat. Commun.

Some minor issues:

1) The references for the FNMR (or if not referenced) should be mentioned in the General Information of the supporting information.

Answer: We thank this referee for his/her suggestion. We have inserted a sentence explaining the references of FNMR in the General Information of the SI.

2) The authors mentioned in the main text that “some of the fluorinated product can only be separated without decomposition with 2% triethylamine in the eluent”. Yet, none the eluent contains triethylamine in the supporting information. The authors should clarify this point in the SI so that the separation could be reproduced in other labs.

Answer: We thank this referee for his/her suggestion. We have provided additional information for all examples that need to be clarified in the SI, for example compounds **23-27, **32-38**.**

3) The authors have polished the language in the main text, there are still many grammatical issues in the supporting information.

Answer: We thank this referee for his/her suggestion. We read the manuscript again and made some additional corrections on the revised version.

Reviewer #2 (Remarks to the Author):

The current paper discloses a useful fluoroalkylating reagent 3,3-difluoropropen-1-yl ammonium salts (DFPAs) for preparation of gem-difluoroallylated compounds. This reagent can be readily prepared from industry chemical 3,3,3-trifluoropropene and allows gem-difluoroallylation of a variety of O-, N-, S-, Se- nucleophiles. The resulting products can serve as a versatile building block for diversified synthesis of fluorinated compounds of medicinal interest. However, the C-nucleophiles seem to be improved their substrate scope. In the current process, only one specific substrate diphenylacetonitrile was applicable to the reaction. Reactions of ketones with beta-H resulted in gem-difluoroallyl vinyl ethers. Although the SN2' reaction used in this paper is well established, the reagent enables to access a variety of gem-difluoroallylated compounds that are not easy to prepare via transition-metal free conditions. Overall, this is a nice paper, I recommend publication of the paper

and suggest minor changes:

1) "difluoroallylation" should be "gem-difluoroallylation".

Answer: We thank this referee for his/her careful reading. We have corrected the mentioned mistake in the revised manuscripts.

2) Explanations should be provided for the formation of gem-difluoroallyl vinyl ethers with ketones bearing beta-H as the substrates.

Answer: We thank this referee for his/her careful reading. We have inserted a sentence to explain the high oxygen regioselectivity in the revised manuscript.

3) Cite the following papers for the direct introduction of difluoroalkyl groups forming X-CF₂-R: Acc. Chem. Res. 2018, 51, 2264; Nat. Chem. 2017, 9, 918; Nat. Chem. 2019, 11, 948.

Answer: We thank this referee for his/her careful reading. We have cited the three references in Ref 31-33.